# A Unified Internal Flow Model with Fluid Momentum for General Application in Shipflooding and Beyond

Riaan van 't Veer *, Joris van den Berg and Sander Boonstra 

Maritime Research Institute Netherlands (MARIN), 6708 PM Wageningen, The Netherlands
* Correspondence: r.vantveer@marin.nl

**Abstract:** This paper deals with the development and validation of a unified internal flow model (UIF) for the prediction of fluid behaviour in a network of 3D floodable cells such as an internal ship compartment subdivisions. The UIF model is incorporated in a generic time-domain ship-motion simulation environment. The flow model uses cell-averaged fluid momentum to account for dynamic (inertia) effects in compartments. A compartment is typically given the size of a ship compartment on board. The fluid solver can keep track of the air pressure in and air flow through compartments under isothermal assumption. Simulation results are compared against experimental data from four different configurations: a single tank draining experiment, a two compartment down-flooding experiment, an oscillating water column experiment under pressure, and a transient flooding experiment with a cruise ship in calm water. The general validity of the UIF model is demonstrated through these four examples. The newly developed UIF model overcomes the shortcomings seen in the steady Bernoulli-based simulations. Its application leads to a more accurate prediction of the floodwater progression in the ship, and it captures the fluid dynamics seen in the oscillating water column device very well, which is not possible using the steady Bernoulli approach. The general applicability of the UIF solver for internal fluid response in maritime application is thereby demonstrated. The effect of the internal compartmentalisation of a cruise ship and the effect of viscous roll damping on the transient roll response is discussed.

**Keywords:** ship damage stability; numerical simulation; unsteady Bernoulli equation; model tests

## 1. Introduction

In this paper, the outline of a Unified Internal Flow model (UIF) is presented that interacts with the time-domain ship-motion solver to predict e.g., the motions of a damaged cruise ship. The UIF solver aims to provide a generic simulation model for all kinds of fluid behaviour in maritime applications. Examples include slow ship ballasting, transient and progressive shipflooding, U-tanks for roll reduction, or moonpool pumping. In some of these problems, fluid inertia effects are dominant; in others they are less important or not significant at all. The basic idea behind the development of the UIF model is that the fluid inertia aspects are implicitly included in the equations. If the flooding geometry and fluid condition are such that fluid inertia aspects are relevant, it will be picked up by the solver, and if not, the steady (quasi-stationary) character of the flow through the floodable geometry is obtained.

A first version of the UIF model was presented in [1], and some examples were given. In the present paper, the model is presented in more detail and further results are shown, including an example of resonant fluid behaviour in a configuration with entrapped air and an example showing the transient flooding of a damaged cruise ship as a model tested at MARIN for the EU-research project Flooding Accident Response (FLARE). The latter experiment is presented and discussed in detail in [2].

The 3D geometry through which the water can flow is called the XHL network (Extensible Hydraulic Library), which is part of MARIN's time-domain simulation environment,

which is called XMF. The XHL network has two modus operandi upon user selection: with (UIF) or without (TOR) accounting for fluid inertia aspects. Both models work with the same input definition apart from one single keyword to set the modus operandus. The TOR model is named after Torricelli who, in 1644, was the first to present a hydraulic theory on the efflux of flow from a container, now known as the Torricelli theorem. It relates the velocity $v_A$ through a submerged orifice to the water height $h$ above the orifice: $v_A = \sqrt{2gh}$. The largest velocity is thus obtained at the initiation of the breach in the tank, when the water height is the largest. In reality, the fluid velocity will (quickly) build up over time before reaching the steady state velocity given by the theorem. The Torricelli efflux velocity can be derived from the steady Bernoulli Equation (1738), used in many simulations tools for dynamic ship stability, including MARIN's FREDYN–XMF solver [3].

The unsteady Bernoulli equation, shown in Equation (1), follows from the Euler equation when applied along a streamline. It involves a time-dependent term that expresses the rate of change in fluid momentum along the streamline between point 1 and 2:

$$\int_1^2 \frac{\partial v}{\partial t} dS + \frac{p_2}{\rho} + \frac{1}{2}v_2^2 + gz_2 = \frac{p_1}{\rho} + \frac{1}{2}v_1^2 + gz_1 \tag{1}$$

The steady Bernoulli equation is obtained when the time-dependent (fluid inertia) term is neglected. In some particular configurations, the unsteady term can be written out (or approximated) as part of the geometrical properties of the configuration. This then leads to a specific set of equations valid for that particular case with a particular flooded state (fluid levels). An example of this can be found in [4,5] for an oscillating water column device (OWC). The experiment with the OWC device includes entrapped air under high pressure, and the case is studied in the present paper.

To overcome some limitations of the steady Bernoulli equation, Lee [6] derived a dynamic orifice equation (DOE). In that derivation, a control volume around the orifice is utilised. The mass of fluid that moves proportionally with the fluid velocity at the orifice opening depends on the opening geometry and the assumed control volume. The DOE leads to a more realistic fluid (surface) behaviour near equilibrium, showing a decaying free surface difference over time. However, it is not expected that application of the DOE will, e.g., predict the natural period of an oscillating water column accurately, since the tank geometry itself is not used.

The UIF model in the present paper constructs a network of what are called virtual pipes. These pipes represent streamlines, and they are constructed between the free surface of the cell and the centroid of the openings through which water enters or leaves the cell. Multiple openings in one cell lead to multiple virtual pipes of different length, diameter, and orientation. The tank geometry is thus part of the equations. The linear momentum change in each cell is updated by the fluid solver and distributed over the different virtual pipes by their geometrical properties in the cell, as is explained later. The virtual flow through pipes mimics the fluid inertia in the network cells, approximately, but also accounts for the actual flooded cell state.

The outline of the paper is as follows. The theory behind the latest developments of the UIF model are presented in Section 2. Section 3 presents a number of simulations in comparison to experimental data found in literature or generated in MARIN research. Final discussion and conclusions are in Section 4.

## 2. A Unified Internal Flow Model Including Fluid Inertia

### 2.1. Introduction

A series of compartments are configured in the XHL network by means of 3D floodable cells and 2D connected openings. A configuration of any complexity can be used. The XHL network can be placed inside a moving ship or evaluated as a stand-alone configuration. The cells with fluid inside and wetted openings are part of the active network for which fluid equations are stated.

The overall idea of the UIF solver is to keep track of the mass ($m$) and the cell-averaged linear momentum ($m\vec{v}$) in the active network. Cells typically have the dimension of a ship compartment. Linear momentum has direction; thus, $\vec{v}$ is the cell-averaged velocity vector with x-y-z components.

The cell-averaged linear momentum is the sum of the 1D fluid momentum through so-called virtual pipes in the cell which basically mimic fluid streamlines. Virtual pipes are modelled between the cell free surface and the wetted openings in that cell. Each virtual pipe has, thus, a 3D orientation given by the centroid of the connections, which are called nodes. The length $l_{\text{pipe}}$ of a virtual pipe is the vector length $|\Delta\vec{x}|$ between the node coordinates. The assumed pipe area is $A_{\text{pipe}} = \sqrt{A_L A_R}$, where $A_L$ and $A_R$ are the wetted node areas. Completely filled cells do not have a free surface. To allow mass transport through such cells, virtual pipes are, in that case, modelled between the cell openings. The number of virtual pipes per cell is governed by the number of openings in the cells, which can be any number.

A linear system $A\vec{y} = \vec{b}$ is formulated that solves the change in (1D) virtual pipe velocities in the UIF network. The forcing on a virtual pipe comes from the pressure difference over the pipe nodes, as is explained later.

In the Torricelli approach (TOR), the virtual pipes are not used, and the flow behaviour is driven by the pressure difference over wet–dry or wet–wet (submerged) openings between cells. The velocity through the opening is determined by the steady Bernoulli equation, assuming known fluid pressures near the opening. In the TOR model, the solution vector $\vec{y}$ is the fluid velocity through all openings.

### 2.2. Construction of Virtual Pipes

As mentioned, the XHL network contains 3D floodable cell volumes and 2D openings between them. An opening has two sides, since it connects two cells: a left side and a right side. The wetted surface of the left and right side of the opening are not necessarily the same in numerical sense. If the free surface of a cell is below the opening, the opening on that cell side is dry, while it can be wet on the other side. If an opening is wet, a node is positioned at the centroid of the surface of the wetted opening side. There is, as well, a node positioned in the centroid of the cell's free surface.

There are two different node types. A wet–wet area between two cells leads to a connected node. A dry–wet area between two cells leads to a free node on the wetted side, in the centroid of that wetted area. The node on the free surface is always a free node. In Figure 1, an example is shown of how the active network can change over time. If there are no wetted openings in a cell (stage 1, right cell), there are no virtual pipes in that cell and the free surface node is not used. In stage 2 of Figure 1, there is a wet–wet opening with a connected node, and a free node in the same opening due to the single-sided wet part of that opening. In stage 3, the opening is fully submerged on both sides, and there is only a connected node in the opening between the two cells.

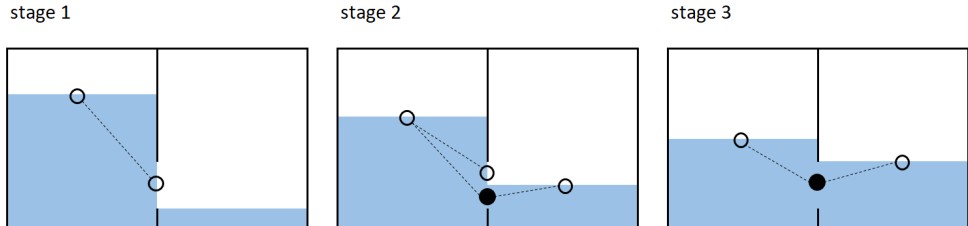

**Figure 1.** Example of network development over time. The connected nodes are shown as solid circles, the free nodes as open circles, and the virtual pipes as dashed lines.

When a cell has multiple openings, multiple virtual pipes can exist that all connect to the free surface node. If a cell is completely full, there is no free surface. In that situation,

there are no virtual pipes towards the (non-existing) free surface, but, to allow mass transport through the cell, there are virtual pipes between submerged openings.

Virtual pipes are only created inside cells, which implies that there is no virtual pipe from a node that connects the network with the outside world. To ensure that there is an inflow to the network from the outside, the Torricelli velocity is used between submerged nodes and the outside world.

*2.3. Solver Description*

The linear system $A\vec{y} = \vec{b}$ that solves the virtual pipe velocities in the UIF modus operandus is a per-cell system with 3 rows (x, y, and z component of the cell-averaged momentum) and $n$ virtual pipe columns. The right-hand side vector $\vec{b}$ is the cell-averaged momentum vector. The solution vector $\vec{y}$ is the set of virtual pipe velocities. The time-stepping solver ensures that the sum of the initial pipe velocity estimate and the change in velocity during the time step satisfies mass conservation. In summary, at each time step, the following actions are taken:

- Determine, for each cell, the centroid of the free surface area and the centroids of wetted openings;
- Construct the virtual pipes in each cell;
- Initialize the virtual pipe flow (average) velocity based on the cell-averaged momentum—see Equation (9);
- Solve the linear system $A\vec{y} = \vec{b}$ for the change in the pipe flow velocities based on the pressure differences over each opening, as expressed in Equation (6);
- Check the solution against the constraints of the cell capacity (full or empty) and, if needed, exchange free nodes for connected nodes;
- Update the cell fluid volumes and the cell-averaged momentum through Equation (9).

*2.4. Solver Equations, Fluid Forcing, and Change of Momentum*

If a simulation starts with an empty network (empty cells), water ingress can only take place when an opening becomes (partially) submerged on the outside. The dynamic pressure at such an opening is as follows:

$$p = \rho g h + \frac{1}{2}\rho v^2_{\text{opening}} + p_{ambient} \tag{2}$$

When there are no waves, the hydrostatic pressure is retrieved from the above equation.

The initial virtual pipe velocity $v_0$ is zero or obtained from the previous time step. The pipe velocity $v$ at the end of the time step $\Delta t$ is as shown:

$$v(t + \Delta t) = v_0(t) + \delta v \tag{3}$$

There are two sets of equations to determine the pipe velocities: (a) mass conservation dictates that the combined velocities conserve mass in the system, and (b) momentum conservation dictates that the change in velocity $\delta v$ is driven by the pressure difference between the virtual pipe nodes.

Newton's second law states that the change in linear momentum equals the sum of external forces $F$ acting on the control volume. The control volume is the virtual pipe. The mass in the virtual pipe is constant during one time step, since the coordinates of the pipe nodes do not change within a time step. The change of linear momentum in the pipe is given as follows:

$$F = \frac{\mathrm{d}}{\mathrm{d}t}(mv) = m\frac{\mathrm{d}}{\mathrm{d}t}(v) = \rho V \frac{\mathrm{d}}{\mathrm{d}t}(v) \approx \rho A_{\text{pipe}} l_{\text{pipe}} \frac{\delta v}{\Delta t} \tag{4}$$

where $A_{\text{pipe}}$ is the pipe area and $l_{\text{pipe}}$ is the pipe length, as defined earlier. Note that the $\mathrm{d}(v)/\mathrm{d}t$ term correlates to the unsteady term in the Bernoulli formulation, Equation (1).

The external forcing on the pipe follows from the pressure difference between the pipe nodes:

$$F = \Delta p A_{\text{pipe}} = (p_L - p_R) A_{\text{pipe}} \tag{5}$$

where $p_L$ is the dynamic pressure at the left node, and $p_R$ the dynamic pressure at the right node of the virtual pipe. Combination of Equations (4) and (5) leads to the following:

$$l_{\text{pipe}} \delta v = \Delta t \left( \frac{p_L}{\rho} - \frac{p_R}{\rho} \right) \tag{6}$$

As observed in Equation (6), the change in velocity $\delta v$ in the pipe is independent from the defined pipe area. It depends on the pipe length, and, hence, on the geometrical properties and fluid state condition of the cell. The length of the virtual pipe is a measure of the fluid mass in the cell that is accelerated or decelerated, i.e., the fluid inertia.

The dynamic pressure in a free node is given by Equation (2). At the free surface, the velocity is zero and the ambient pressure is retrieved for those free nodes. For (partially) submerged openings, the opening velocity in Equation (2) is related to the pipe velocity through the area ratio of the pipe and the opening:

$$v_{\text{opening}} = \frac{A_{\text{pipe}}}{A_{\text{node}}} (v_0 + \delta v) = \alpha (v_0 + \delta v) \tag{7}$$

The orientation of the pipe with respect to the opening node is not accounted for in the present UIF model. It is assumed that the flow velocity is normal to the opening.

Next, Equation (7) is used in Equation (2) and linearised under the assumption that the velocity change $\delta v$ over a small time step is small:

$$p/\rho = gh + \frac{1}{2}(\alpha v_0 + \alpha \delta v)^2 + p_{\text{amb}}/\rho \approx gh + \frac{1}{2}(\alpha v_0)^2 + \alpha^2 v_0 \delta v + p_{\text{amb}}/\rho \tag{8}$$

The hydrostatic part $gh$ and the pure velocity term $v_0$ are known quantities and, thus, included in vector $\vec{b}$ of the solver. The linearised contribution with $\delta v$ is included in the matrix $A$ on the left hand side. The ambient pressure is known for fully vented compartments, or needs to be solved when entrapped air is accounted for.

At connected nodes, the fluid pressure is unknown. These nodes are not an endpoint of a Bernoulli streamline but exist somewhere in the middle. Thus, the linear system needs to solve the pressures at connected nodes together with the change in pipe velocity $\delta v$. For each $\delta v$ component in $\vec{y}$, a momentum conservation equation is used, and for the pressures $p$ at connected nodes in $\vec{y}$, mass conservation applies. This leads to a determined system.

The mass in a cell is updated at the end of the time step given the net sum of the mass transport to the cell. The averaged linear momentum in a cell is given by the following:

$$m_{\text{cell}} \vec{v}_{\text{cell}} = \rho V_{\text{cell}} \vec{v}_{\text{cell}} = \sum \rho A_{\text{pipe}} \Delta \vec{x} v_{\text{pipe}} \tag{9}$$

where $V_{\text{cell}}$ is the volume of the cell and $\vec{v}_{\text{cell}}$ is the vector velocity in the cell with $v_x, v_y, v_z$ components. The orientation of the pipe is given by the vector $\Delta \vec{x}$ between the pipe nodes.

### 2.5. Energy Losses over an Opening

When a fluid flows from one cell to another through an opening, energy losses take place. The amount of energy loss depends on the local geometry of the opening, and it is modelled by the hydraulic resistance coefficient $\zeta$. An extensive collection of coefficients for a large range of different flow sections is given in [7]. It is mentioned that some coefficients are accurate and based on experiments, while other coefficients are more indicative. Resistance coefficients depend, in general, on the Reynolds number as well.

The pressure drop over the opening is modelled as shown:

$$\Delta p(v) = \frac{1}{2}\rho\zeta v|v| \tag{10}$$

The resistance coefficient $\zeta$ can be included in the dynamic pressure Equation (8). As can be derived from the Torricelli theorem, the discharge coefficient $C_D$ and resistance coefficient $\zeta$ are related: $\zeta = 1/C_D{}^2 + 1$. The discharge coefficients are typically input by the user to the simulation setup.

The most common opening between ship compartments is a rectangular-shaped door-opening or hatch in a wall. This opening is denoted as an orifice. Flow separation is most likely to occur on the outflow side. When the orifice is partly submerged, disturbances on the free surface can be seen and a mixture of air and water is likely to complicate the flow behaviour. All these effects are captured by the resistance coefficient. As discussed in [7], the flow resistance depends on the Reynolds number, which typically varies over time due to the change in fluid velocity through the opening. When $Re > 1 \times 10^5$, the (submerged) orifice resistance coefficient is, according to [7], nearly constant, and $C_D \approx 0.59$. The Reynolds effects during discharge are addressed in an example in the present paper.

*2.6. Entrapped Air*

In many (but not all) flooding scenarios, the entrapped air exchange between compartments amounts to air bubbles passing through well-mixed water. In that case, the heat exchange between the air and the water is efficient. Since water has a high thermal capacity, it is valid to assume that the temperature remains constant and that the air temperature equals the water temperature. The process is isothermal. When there is venting of air from a large entrapped air volume through a small opening, there is fast expansion of the air and, most likely, negligible heat exchange, so that the process is isothermal as well. The flooding of a ship on model scale or full scale is considered to follow an isothermal process.

Adiabatic processes typically take place in systems with fast-changing pressures that are isolated from the surrounding. If they are to be simulated, a temperature equation is to be written out, and this significantly complicates the implementation.

The current implementation of the air entrapment functionality follows the isothermal process. The more complicated adiabatic process functionality is under consideration but not yet implemented. Both the TOR and the UIF model include the same air modelling.

The gas law for an isentropic ideal gas is given below:

$$p_a V_a{}^\gamma = c \tag{11}$$

where the heat capacity ratio of air $\gamma$ is 1.0 for an isothermal process and about 1.4 for an adiabatic process. The gas volume $V_a$ has an absolute pressure $p_a$. The volume of a cell equals the gas volume plus fluid volume.

The pressure and the air volume are monitored in each cell. To keep track of the amount of entrapped air, the cell state $c = p_a V_a$ is used. Knowing $c$, the static air pressure can be calculated. The stationary Bernoulli equation is used to calculate the air velocity $v_a$ knowing the pressures at the left and right side of an opening. The volumetric rate of air flow $Q_a$ is obtained by multiplying the velocity with the cross sectional area through which air can flow. Mass conservation then requires $\dot{c}_L = -\dot{c}_R$.

The current implementation couples the fluid solver and the pressure equations to obtain a consistent solution within one time step. This (rather complex) numerical implementation is not further discussed herein, as further research is ongoing. Air typically stiffens the solution system, so that a smaller simulation time step is often required to obtain a time-step-converged solution. The air accumulator model, as described in [6], has been implemented as well; it works well with our model, but it needs further verification to establish its general use. The example with high air pressures in the present paper is calculated with a small time step and without using the air accumulator functionality.

## 3. Applications and Validation

### 3.1. A Single Tank Outflow Experiment to Measure the Discharge Coefficient

In the EU project Flooding Accident Response (FLARE) ), an experiment was conducted to establish an average discharge coefficient for the typical openings used between compartments in model experiments. The obtained discharge coefficient was used in the FLARE benchmark study [2] to obtain a fair comparison between different simulation codes applied.

The configuration of the experiment is shown in Figure 2. The rectangular box internal space measures L × B × H of 300 × 450 × 233 mm. A sharp-edged opening of $b = 43$ mm and $h = 34$ mm was fitted at 55 mm from the bottom of the tank. The wall thickness of the orifice was 2 mm. The wall thickness of the box is not shown in the figure. The box was vented by an opening at the top of 7.0 cm$^2$. At the start of the simulation, the compartment was completely full. A vertical sliding door was used to open the box quickly. The water level was measured at two locations (corners) in the box at 200 Hz, showing nearly identical results.

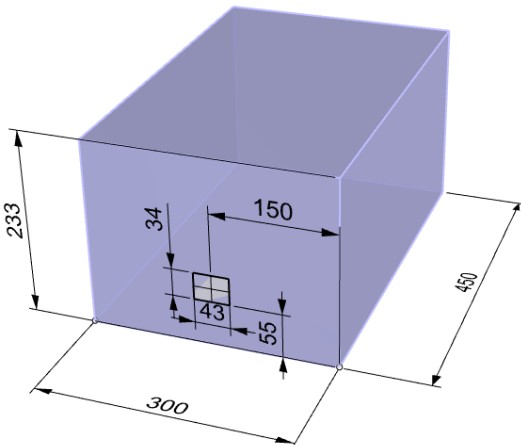

**Figure 2.** A 3D view of the outflow compartment. Dimensions in mm.

The water height measured at one sensor (given w.r.t. orifice base) and the low-pass filter-averaged water height (given w.r.t. compartment base) are shown in Figure 3. As observed, the compartment does not drain to exactly the orifice lower edge (55 mm above base); a small meniscus of about 2.5 mm water remains at the opening.

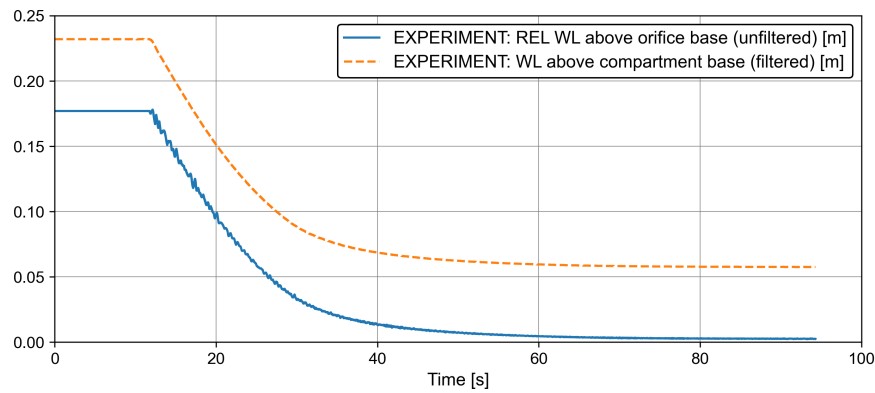

**Figure 3.** Flow level in the compartment during discharge.

The flow rate $Q$ through the opening should match the flow rate calculated from the water height surface decline over time. The discharge through the opening can be calculated, assuming a steady state flow:

$$Q = \frac{2}{3}\sqrt{2g}C_D b(H_b^{3/2} - H_t^{3/2}) \tag{12}$$

where $H_t$ is the water height above the top edge of the orifice and $H_b$ is the water height above the bottom edge. When the water falls below the top edge, $H_t$ is zero. Using Equation (12), the discharge coefficient $C_D$ in the experiment was derived. From the flow rate through the opening and its wetted surface $A_{\text{wet}}$, an average discharge velocity $v_a$ and a Reynolds number $Re = (v_a\sqrt{A_{\text{wet}}})/\nu$ were calculated. The results are shown in Figure 4.

As observed, the $C_D$ coefficient for the orifice flow in the model experiment varies with the Reynolds number. Based on data found in reference [7], a set of equations is derived to capture the Reynolds dependent discharge through an orifice—see Equation (13). These equations are implemented in the UIF/TOR solver to provide a convenient user option avoiding any further discussion on, e.g., scale effects. A fair correlation is observed between the measured and theoretical $C_D$ from Equation (13).

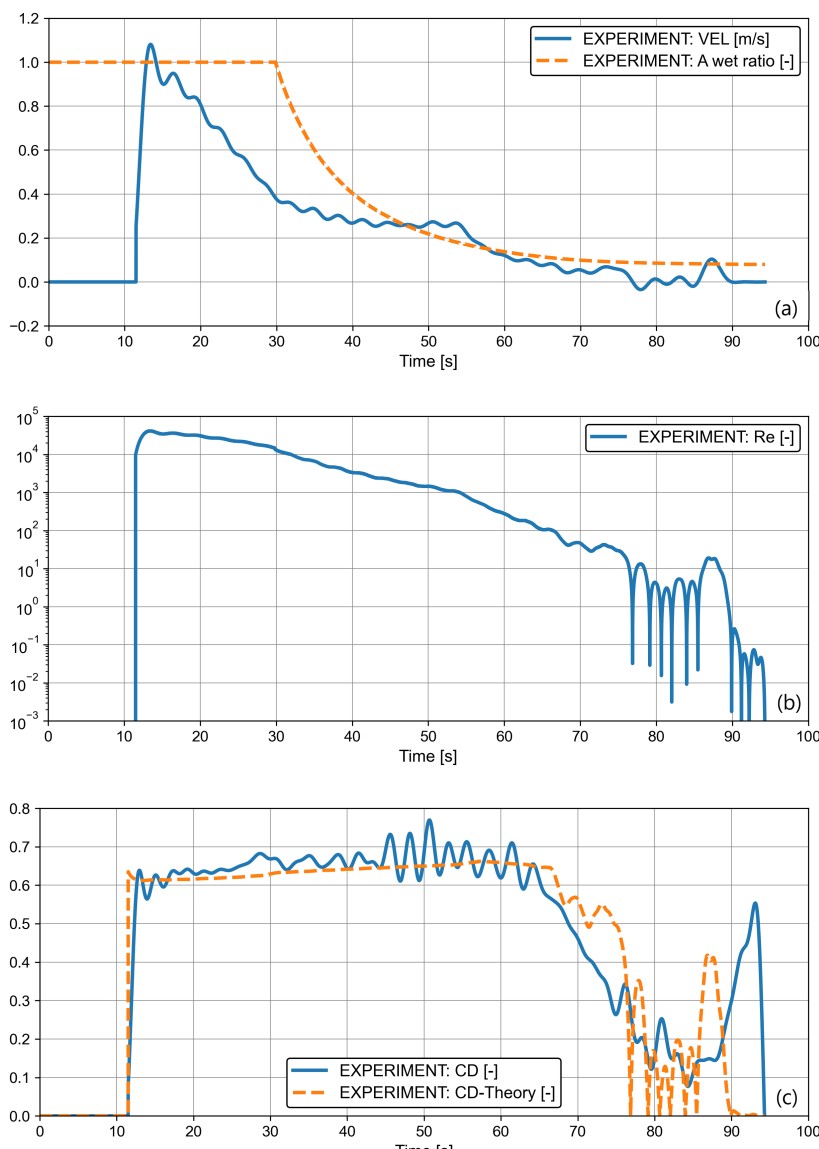

**Figure 4.** Experimental results. (**a**) Velocity through the orifice and normalized wetted area of the orifice, (**b**) derived Reynolds number, and (**c**) measured and theoretical discharge coefficient over time.

$$C_D \approx \begin{cases} Re/(19.82 + 1.35Re) & \text{if} \quad Re \leq 100 \\ Re/(5.0 + 1.5Re) & \text{if} \quad 100 < Re \leq 500 \\ 0.59 + 0.20/Re^{1/6} & \text{if} \quad 500 < Re \leq 1E4 \\ 0.59 + 4.5/\sqrt{Re} & \text{if} \quad Re \geq 1E4 \end{cases} \tag{13}$$

In Figure 5, the comparison is shown between the experiments and the UIF and TOR simulations. In both simulations, the Reynolds-dependent discharge coefficient was used. The numerical results compare very well to the experiment. In the UIF model, the discharge velocity builds-up over time, demonstrating the effect of fluid inertia. In the TOR model, the maximum velocity is obtained at the second time step, and not at the first time step, due to Equation (13).

The UIF model demonstrates a more physical start-up phase of the flow, but the steady velocity is quickly reached and there is hardly any benefit from the UIF model compared to the TOR model. This single-tank discharge experiment can be well captured by both models. It would require a more detailed and accurate experiment to conclude if the inertia start-up effect is well captured by the UIF model. The Re dependency in the orifice flow is, however, clearly demonstrated and confirmed by theory.

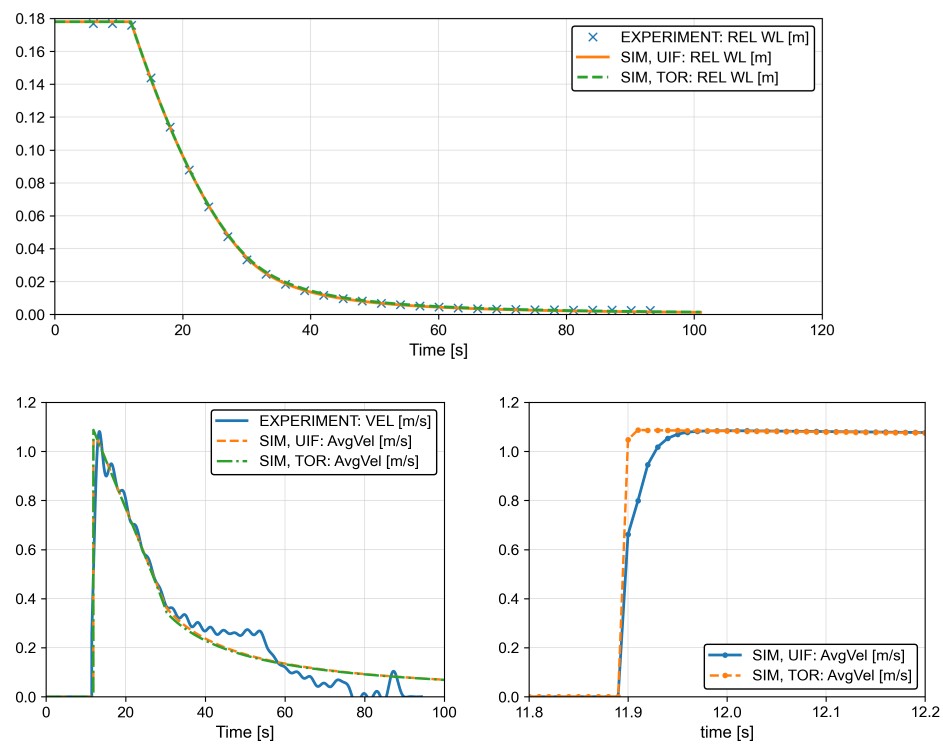

**Figure 5. Top**: Comparison of water height *H* with respect to the orifice bottom edge, and **Bottom**: Average discharge velocity.

### 3.2. A Two Compartment Down-Flooding Experiment

Within the EU FLARE project, a two-compartment down-flooding experiment was conducted with the configuration shown in Figure 6. The draft of the setup was 400 mm, measured from the inner bottom of the lower compartment. An air pipe with a diameter of 50 mm was used to vent the compartments. In each compartment, wave probes were positioned close to the corners to measure the water level, as indicated in the setup. The experimental data can be found at http://shipstab.org/index.php/data-access (accessed on 26 April 2023).

A ComFLOW simulation of the down-flooding experiment was performed. ComFLOW is a Cartesian (cut cell) grid-based Volume of Fluid CFD solver, using a staggered

finite-volume discretisation of the Navier-Stokes equations [8]. Automatic grid refinement is used by means of a surface and object tracking criterion. Assuming fully vented compartments, the ComFLOW simulations were executed in single-phase mode, solving only the computational cells which contained liquid.

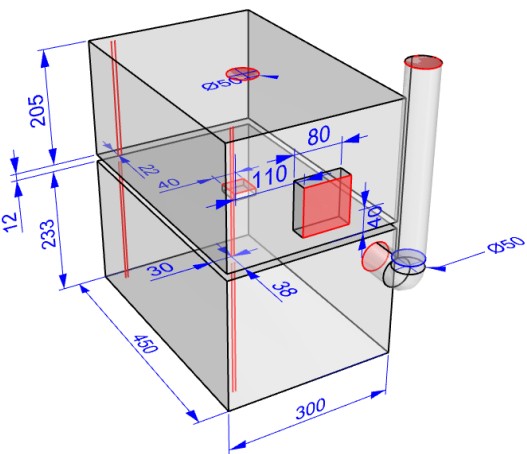

**Figure 6.** Two-compartment down-flooding experimental setup. The downflood opening, positioned in the middle of the deck, measured 40 × 40 mm. The breach in the upper compartment measured 80 × 80 mm.

Several visualisations of the compartment filling level and flow velocities, as obtained by ComFLOW, are shown in Figure 7. The 2D snapshots are taken in the middle of the breach. A clear fluid jet is observed to start at the breach and hit the deck just around the opening. This effects the flow velocity through the downflood opening. The free surface in the upper compartment shows significant spatial height variations at the start of the flooding. Near the end of the simulation, the vent pipe fills with water, and some oscillations are observed in the free surface level in the pipe. The snapshots demonstrate that, while the experimental setup perhaps looks simple, the flow pattern is far from simple. Obviously, most of the (local) details cannot be captured by the UIF or TOR model, in which each compartment is modelled as a single cell with an assumed horizontal free surface.

In Figure 8, the measured water height in the upper compartment (Rel 17) and in the lower compartment (Rel 25) are shown. The ComFLOW results compare very well to the experimental data, and similar free surface oscillations on the wave probes are found as measured. The TOR and UIF results are obtained using a calibrated discharge coefficient of 0.80 for the downflood opening. The TOR model then gives a very good prediction of the water level in both compartments. Using the UIF model, the correlation to the experiments in the upper compartment is slightly less good. There is a small increase in the water level compared to the experimental data, which points to the inflow velocity being too high at the breach. This is attributed to the connected node and requires further study in future research. The calibrated discharge coefficient of 0.80 in the downflood opening is higher than expected, but can be explained based on the CFD visualisation. As such, CFD simulations provide valuable detailed flow information that are difficult to obtain from physical experiments.

In Figure 9, measured water heights are compared to simulation results that account for air entrapment. To obtain stable simulations, the time step is reduced by a factor 100 in the UIF model. As a consequence, the calculation time increases from 30 s to about 30 min for 50 s simulation time. Using the TOR model, the time step could be kept the same for stable simulations. Using the UIF model, an oscillating free surface in the pipe is obtained. Similar oscillations were seen in the ComFLOW result. The TOR model predicts an equilibrium without any overshoot. The flooding of both compartments is similar with or without the air modelling. The only difference is the slightly slower filling of the lower compartment when air is temporarily entrapped near complete filling. The model does

allow for air bubbles to escape through an opening that is fully wet on one side only. That is the case here, since the air entrapment is on top of the fluid in the lower compartment. Thus, the UIF model eventually allows for the bottom compartment to fill up completely. This effect is seen as well in the experiments and CFD results shown in Figure 9, but at a much quicker rate, perhaps initiated by the fluid disturbances.

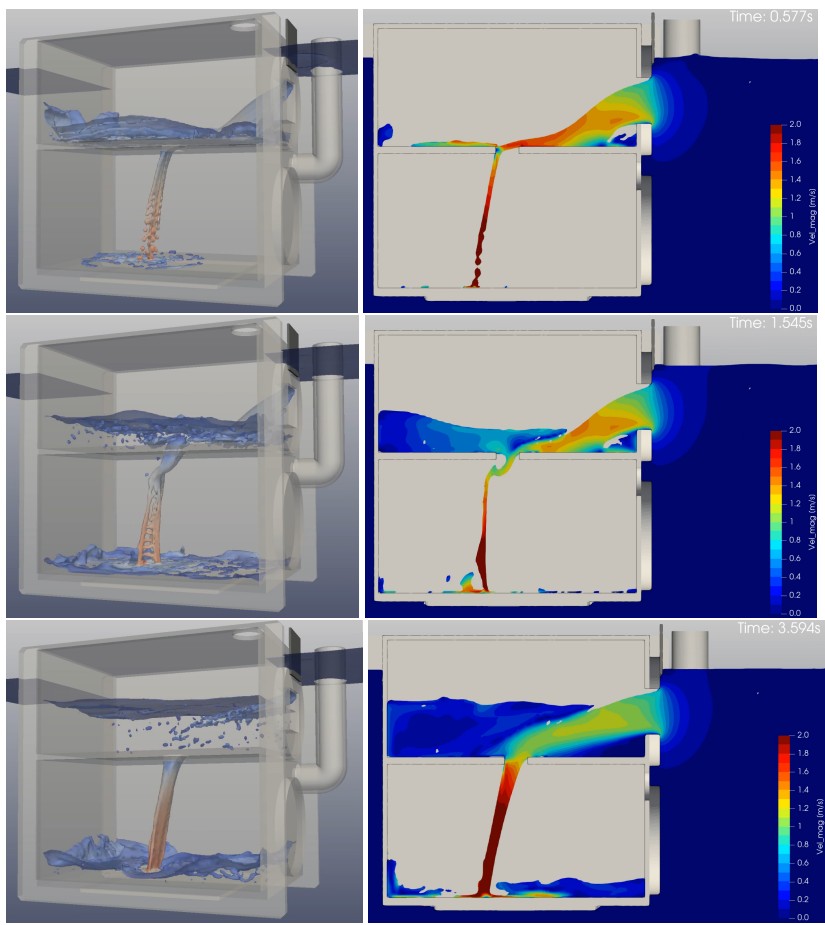

**Figure 7.** ComFLOW CFD results, 3D and 2D view, with model scale velocity. Snapshots at $T = 0.577$, 1.545, and 3.594 s.

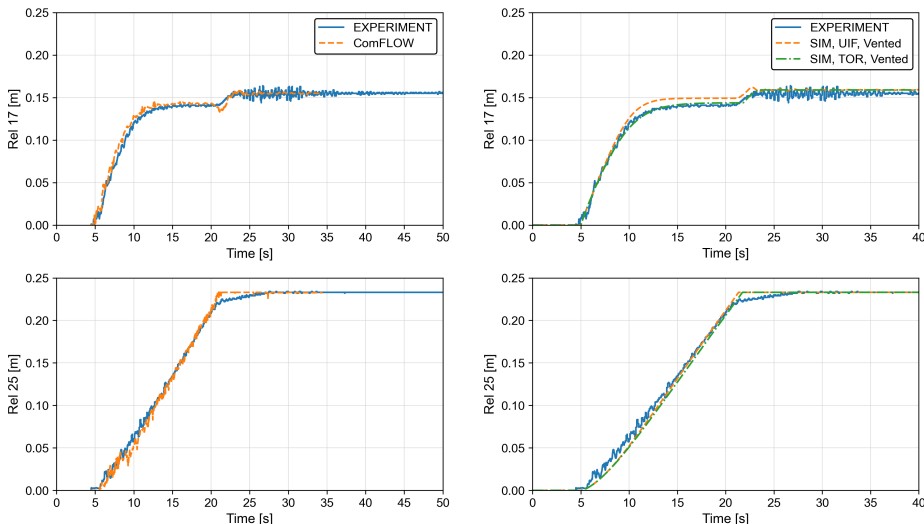

**Figure 8.** Comparison of water height. Experimental, ComFlow (CFD), TOR simulation, and UIF simulation.

Overall, it is concluded that the air modelling for this configuration does not play an important role and that the vent openings were sufficiently large to obtain a fully vented condition. In vented conditions, the simulations are about twice as quick as real time, although the time step could be set even larger to further speed up the simulation time had the small diameter pipe not been part of the model.

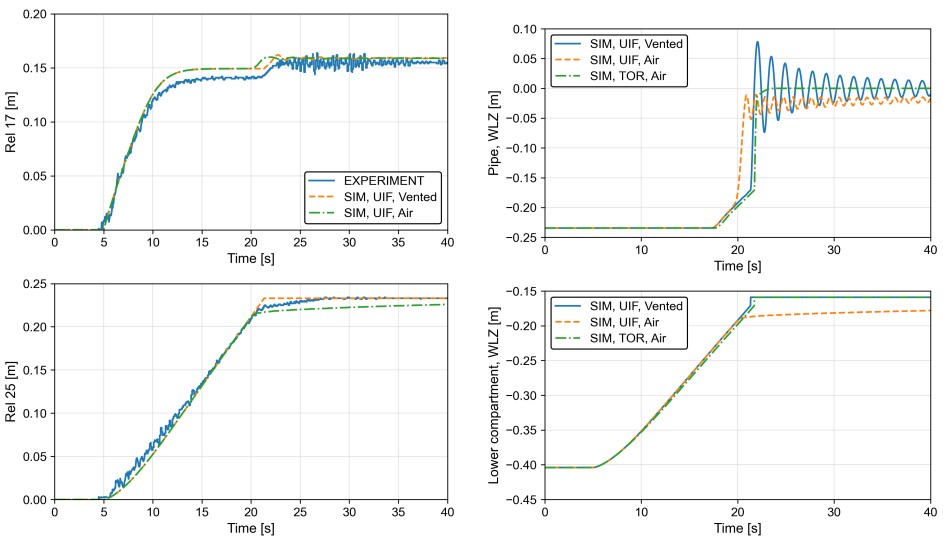

**Figure 9.** Comparison of water heights. Simulations with air modelling.

### 3.3. Behaviour of an Oscillating Water Column (OWC) for Wave Energy Harvesting

The oscillating water column (OWC) is one possible system to harvest wave energy. In [4,5], results are presented from a small-scale experimental device, denoted as REWEC1 (REsonant sea Wave Energy Converter, solution 1). A 2D sketch of the cross section of the configuration is shown in Figure 10. It is essentially a U-tube. The width of the experimental setup is not provided. The 3D configuration used for the analysis in the present paper is given in Figure 10 as well. The width was set to 0.5 m. The other dimensions were Lc = 0.85 m, b = 0.366 m, c = 0.797 m, d = 0.797 m, s = 0.032 m, t = 0.153 m, w = 0.1 m, and h = 0.60 m. The setup consisted of three cells: caisson-1, caisson-2, and the duct. There were two connecting openings and one air vent opening in caisson-1. Mass conservation in the system dictates that $\Delta a/\Delta\zeta = b/d$.

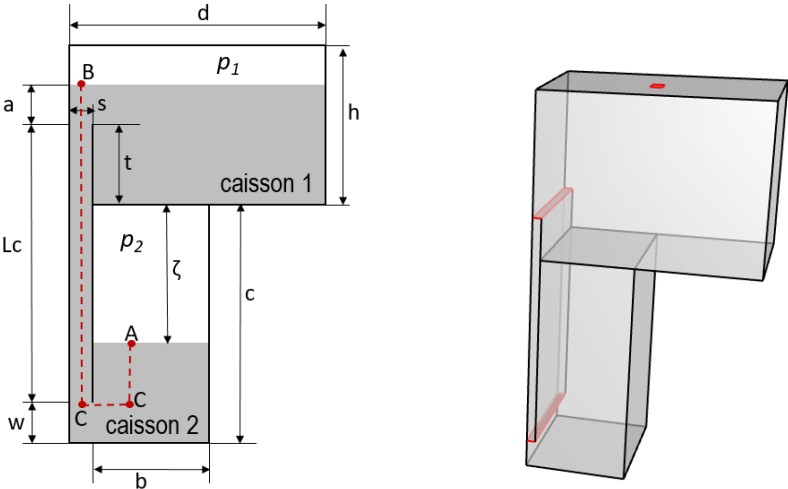

**Figure 10. Left:** 2D cross section of the OWC scale experiment adopted from [4], with the assumed Bernoulli streamline and **Right:** 3D simulation setup utilizing three compartments, two internal openings, and one vent opening at the top (all indicated in red).

The upper caisson is used in order to simulate the REWEC1 submergence, and it is used to initiate the starting condition in the experiment. Each caisson is equipped with a valve to close or open the tank to the atmosphere. At the start, the valve of the lower caisson is closed and water is put into the upper caisson so that the lower caisson is partly filled with an air pocket under pressure. Additional air is then pumped into the lower caisson until the desired resting condition is achieved with regard to atmospheric pressure in caisson-1. Then, the valve of the upper caisson is closed and air is pumped inside it, leading to $p_1 > p_{atm}$ and a further increased pressure in caisson-2. When the valve in caisson-1 is then opened, the pressure in this caisson drops to atmospheric pressure and the free damped harmonic oscillation can occur, converging, in the end, to the equilibrium resting condition. The diameter of the upper valve in caisson-1 is 0.5 inch. Construction details, such as wall thickness of the pipe connecting both caissons and the pipe opening geometry, are not known. Sharp-crested duct openings are, therefore, assumed.

In [5], an analytical model is presented for the system behaviour based on the unsteady Bernoulli equation with the assumed streamline, as indicated in Figure 10. The unsteady term is integrated in two parts: A-to-C and C-to-B. This leads to a non-linear equation, which is linearised to obtain the expression for the undamped natural period of the system:

$$f_n = \frac{1}{2\pi}\sqrt{\frac{C}{A}} = \frac{1}{2\pi}\sqrt{\frac{(\gamma/\zeta_0)[p_{1,0}/\rho + g(\zeta_0 + t + a_0)]}{(c - w - \zeta_0) + (b/s)L_c}} \tag{14}$$

where $\gamma$ is the heat capacity ratio. The other symbols in Equation (14) refer to the drawing shown in Figure 10. The subscript 0 denotes the equilibrium resting condition. As observed in Equation (14), the heat capacity ratio $\gamma$ changes the stiffness of the system by a factor $\gamma$ and, thus, the natural frequency by a factor $\sqrt{\gamma}$. The term between square brackets represents the pressure of the air pocket in caisson-2; hence, this pressure affects the natural period. In the UIF simulations, the heat capacity ratio $\gamma = 1$ is used; in the experiments, an adiabatic behaviour is assumed, with $\gamma = 1.4$.

Out of the four given cases given in [4], case C was randomly selected. The experimental conditions are specified in Table 1. First, the resting condition was verified by specifying the pressure in caisson-2, $a_0$, and $\zeta_0$, and by introducing an open vent to the atmosphere in caisson-1. To obtain an equilibrium pressure of 1077 Pa in caisson-2, the fluid level in caisson-1 had to be set to $a_0 = 0.135$ m, while the reported value in the experiments was $a_0 = 0.079$ m. Using the reported fluid level, the equilibrium condition was reached at $p_2 = 1072$ Pa, which is a difference of only 0.5% compared to the reported value. The fluid level in caisson-2 was then $\zeta_0 = 0.3678$ m. Hence, it is most likely the pressure reading that has the least accuracy.

The dynamic simulation was started using the non-equilibrium condition specified in the experiment—see Table 1. The dynamic simulation result is shown in Figure 11. The natural period in the simulation is 1.22 s, versus 1.0 s in the experiment. This difference is mainly caused by the difference in heat capacity ratio. The major energy losses in the system seem to come from the internal pipe openings, since this is the only damping accounted for in the UIF model, assuming sharp-edged orifice discharge coefficients, as per Equation (13). The main difference in the setup is that the simulations required a slightly larger vent opening to obtain the rapid decrease in the pressure in caisson-1 and the oscillations as presented. The present results are obtained using a vent opening diameter of 1.8 inch. Using a vent with 0.5 inch diameter leads to a slow decay. This requires some further investigation of the UIF modelling. Zero energy loss on the vent opening was assumed.

The end condition in the dynamic simulation is shown in Figure 12. The results are reported in Table 1. Both the UIF and the TOR model predict the same equilibrium end condition, but the dynamics in the system are only captured by the UIF model. As observed, the final fluid level in caisson-1 is 0.028 m higher than the starting value, while it is 0.028 m lower in the experiment. The pressure in the simulation matches the experimental value.

This is most likely a reporting issue, since there is no mass conservation in the reported experimental fluid levels either.

**Table 1.** Initial non-equilibrium starting conditions and resting condition. $p_{atm} = 1013$ Pa.

| Test | $\zeta$ [m] | a [m] | $p_1$ [Pa] | $p_2$ [Pa] |
|---|---|---|---|---|
| Experiment—starting (in) | 0.306 | 0.107 | 1227 | 1288 |
| Experiment—end/resting (0) | 0.366 | 0.079 | 1013 | 1077 |
| Simulation—starting (in) | 0.306 | 0.107 | 1227 | 1288 |
| Simulation—end | 0.366 | 0.135 | 1013 | 1077 |
| Simulation—resting (0) | 0.366 | 0.079 | 1013 | 1072 |

The overall finding is that the UIF model captures the behaviour of the OWC system well. At model scale, the system behaves according to an adiabatic system. At full scale, an isothermal process is considered more likely, and this is what is modelled in the UIF solver. If so, the natural period of the full-scale system can be well predicted. The damping decay rate can be well predicted using the loss of energy at the internal openings. The steady Bernoulli equation model variant (TOR) rightfully predicts the equilibrium resting condition, but, as expected, it cannot predict the dynamic system behaviour.

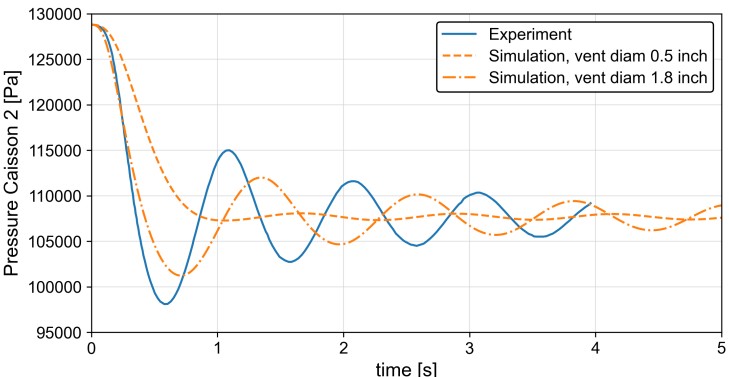

**Figure 11.** Comparison of pressure in caisson-2 between experiment (case C from reference [5]) and simulations.

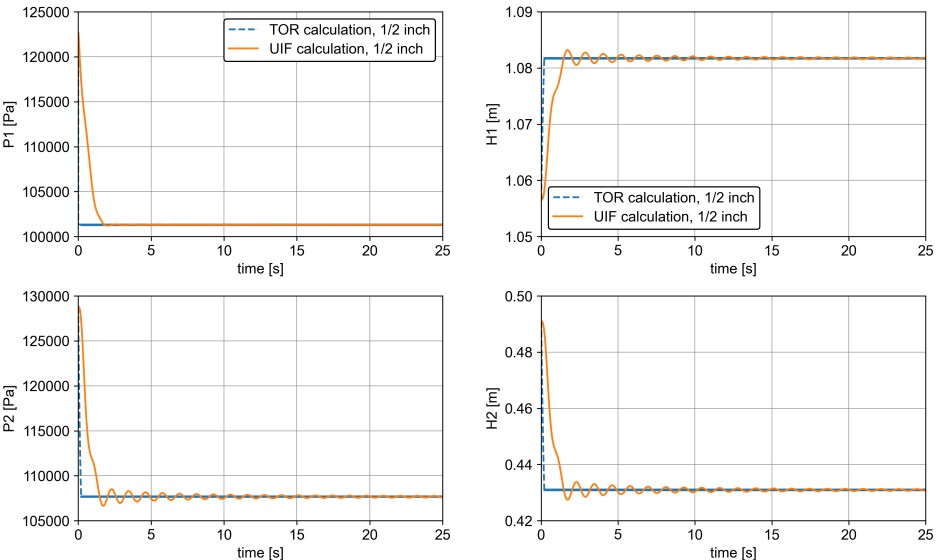

**Figure 12.** UIF and TOR simulation while approaching the resting condition. Water level *H* w.r.t. bottom of caisson-2.

### 3.4. Flooding of a Cruise Ship in Calm Water Conditions

Within the EU FLARE project, MARIN conducted model tests using a 1:60 scale model of a large cruise ship, as shown in Figure 13. The model tests' data are available at http://shipstab.org/index.php/data-access (accessed on 26 April 2023). This concerns an unbuilt large cruise ship (about 95,900 GT), with lines and internal arrangement provided by Chantiers de l'Atlantique to the consortium. The hull lines, the internal subdivisions used in the model tests, and various experimental results can be found in [2]. This reference presents the results of an international benchmark study conducted within the EU FLARE project, showing numerical results of various time domain codes developed for dynamic shipflooding simulations.

The overall length of the ship is about 300 m, with a breadth of 35.2 m. For the model tests, the deepest subdivision draught of 8.20 m was selected, in conjunction with three-compartment damage. The length of this damage is beyond regulations, for the purpose of creating complex and critical damage. It extends from about 134 m to about 180 m forward from APP. According to present regulation, the smallest allowed metacentric height (GM) for this draught is 3.50 m, but for the purpose of the model tests, a lower GM of 2.36 m was selected.

The six-dof XMF time-domain simulations utilized the impulse–response functions calculated from the intact ship hydrodynamics database, obtained from MARIN's 3D panel code SEACAL. The KG was set to 17.45 m to obtain the GM of 2.36 m. The time-varyied hydrostatics and wave loads were calculated on the actual wetted hull of the ship given by the 3D geometry of the ship. Calculations were performed at full scale with a time step of 0.1 s.

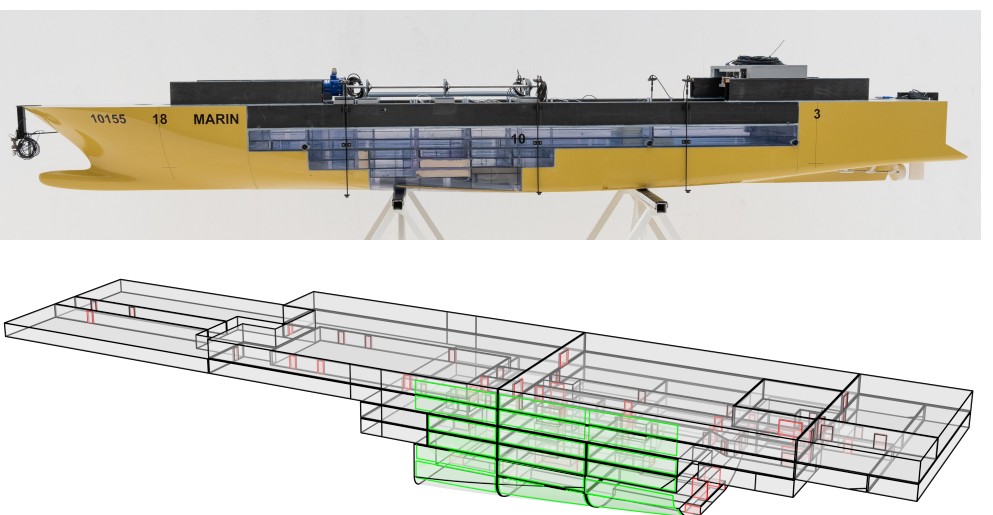

**Figure 13.** Cruise ship side view (non-damaged port-side) and drawing of internal space on all decks. Openings (red) with external breach (green).

The internal subdivision of the damage spans vertically over six decks, up to 20.16 m above base. In the simulation model, all internal compartments are modelled as built on model scale, including the bulkhead and deck thickness of 24 cm (4 mm on model scale). The volume of door-openings in bulkheads is added as an extension to one of the adjacent compartments. The total volume of the intact ship below the water line is 164,300 m$^3$; the volume of the floodable compartments amounts to 48,012 m$^3$.

The floodable space on the lowest deck is shown in Figure 14. Two watertight bulkheads divide the space into three separate compartments that span the width of the ship. Each compartment is subdivided at logical points where the cross section of the tank changes significantly. This results in three tanks along the width of the ship, as shown in Figure 14. Water level probes used in the model experiment are indicated by the Rel

keyword. The aft compartment extends over deck 1 and 2 without an intermediate deck. The ship damage is created on starboard side.

Following the floodwater path from the breached starboard side to the intact port side, the fluid goes through a sudden contraction and expansion due to the narrow centreline tanks. The energy losses are different than for an orifice in a bulkhead. The resistance coefficients used are those associated with typical pipe flows with a sudden contraction or expansion of the pipe diameter (see e.g., reference [7]). For a sudden expansion, the Borda–Carnot energy loss equation ($\Delta E = \xi \frac{1}{2}\rho(v_1 - v_2)^2$) with empirical loss coefficient $\xi \approx 1$ is used, with resistance coefficient:

$$\zeta = (1 - A_{\text{small}}/A_{\text{large}})^2 = 1/C_D^2 - 1 \tag{15}$$

This amounts to a $C_D$ coefficient of about 0.70 to 0.80 for the lower compartments. The simulations are, therefore, carried out with a value of 0.75 for the deck 1 openings, as seen in Figure 14. For all other openings, including the side breach in the ship, Equation (13) is applied.

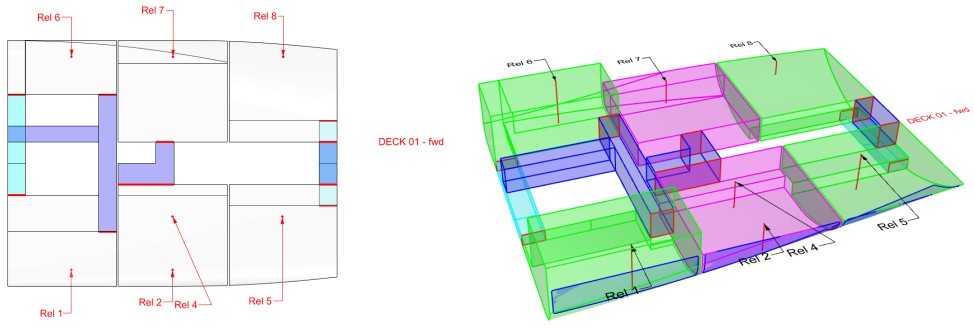

**Figure 14.** Deck 1 subdivision and position of relative wave probes.

The viscous roll damping of the ship is captured by linear and quadratic damping coefficients. The coefficients are tuned to match the free roll decay experiment, as shown in Figure 15. The bilge keel length on the intact side is about 90 m; on the damaged side, it measured 45 m, as it is removed around the external breach opening. The roll radius of gyration was tuned to match the roll period at the start of the decay, resulting in $k_{xx} = 15.5$ m. During the flooding experiments, a horizontal four mooring-line soft-mooring arrangement was used, which was included in the numerical setup as well. Linear and quadratic sway and yaw damping coefficients were tuned to match the sway and yaw decay experiment in calm water condition. These can be seen as low-frequency manoeuvring coefficients.

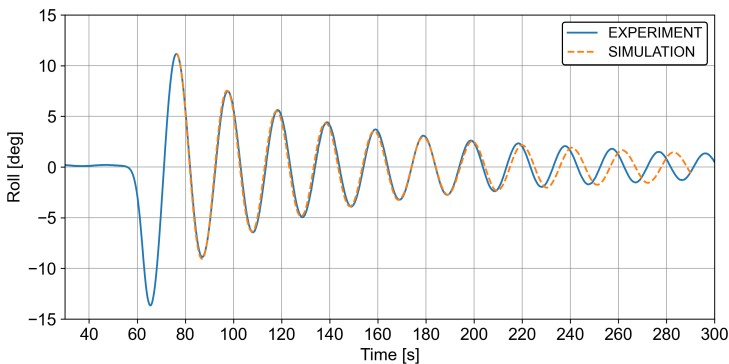

**Figure 15.** Roll decay comparison for cruise ship in calm water.

When the breach in the side is opened in calm water, the ship starts to roll towards the damage as the water egresses in, before reaching a final equilibrium condition. During this motion, the roll damping is expected to be much larger than in the intact condition, since multiple deck edges move through the water. The extent of damage is about 45 m, which is half the length of the intact bilge keels, and about five decks get submerged.

The roll response of the ship in calm water after breaching is shown in Figure 16. In all three simulations, the UIF model was used. The ship response with the subdivision of deck 1 shows a good correlation to the experiment during the transient roll. To obtain this match, the intact roll damping coefficients (linear and quadratic) were multiplied by a factor four. The final equilibrium condition is rather well predicted. With less roll damping, the ship capsises during the first transient roll. The increase in the roll damping can be understood given the fact that multiple deck edges are dragged through the water, creating vortices and, thus, damping. The two simulation results without the deck 1 subdivision show a much lower roll response, demonstrating that using the intact roll damping for the damaged ship leads to a roll peak of about 19 deg. Using the calibrated roll damping in the damaged condition, the roll peak lowers to about 15 deg. This is about half the obtained roll response in the simulation with the subdivision on deck 1. The factors that thus contribute the most to the roll response peak are the amount of roll damping and the degree of subdivision on the lowest deck.

In [2], simulation results of various simulation tools are shown for the present case as part of the international benchmark on flooding conducted in the FLARE project. The subdivision of deck 1 varies in the different tools following practices and rules as considered most applicable by the user. A common approach is that some kind of subdivision is required to develop sufficient roll. There is no information on the applied roll damping. The final equilibrium condition is often well captured, but the transient response differs in accuracy compared to the experiment.

The simulation results for the TOR model are shown in Figure 17. Using the subdivision on deck 1, the ship capsises when the TOR model is applied with the calibrated roll damping for the UIF model. Using roll damping coefficients six times higher than the intact roll damping, the ship survives the first transient roll, but the roll response character is different than in the UIF simulation. Without any subdivision on deck 1, the roll response is much smaller but, in that case, comparable to the UIF simulation results for the same internal configuration.

The present example shows some of the challenges to obtain an accurate prediction of the transient shipflooding response for large damages utilizing the Bernoulli equation. The flow inertia aspects captured by the unsteady UIF model lead to a better correlation with the experimental results compared to the steady TOR model. The UIF model is considered an interesting step forward for dynamic stability simulations when fluid inertia is relevant during the transient flooding phase. Still, the UIF model only approximates the real fluid behaviour. As demonstrated in reference [2], an accurate prediction of the transient ship roll response is far from trivial for any simulation tool. User experience is important to define a proper internal subdivision and model setup, but if that experience is gained e.g., through model test experiences as discussed, realistic (transient) flooding simulations can be achieved. As such, the Bernoulli-based simulation tools can provide viable insights in dynamic ship stability.

The main factors of influence in the presented case are the internal subdivision layout on deck 1 and the roll damping coefficients, which capture the roll damping physics of appendages and the external damage extent. The roll damping of damaged ships was investigated in the FLARE project and the main findings were reported in [9]. One of the conclusions was that the damping depends as well on the ship draught and heel, complicating the roll damping prediction for damaged ships even further. Hence, further research on the topic is required.

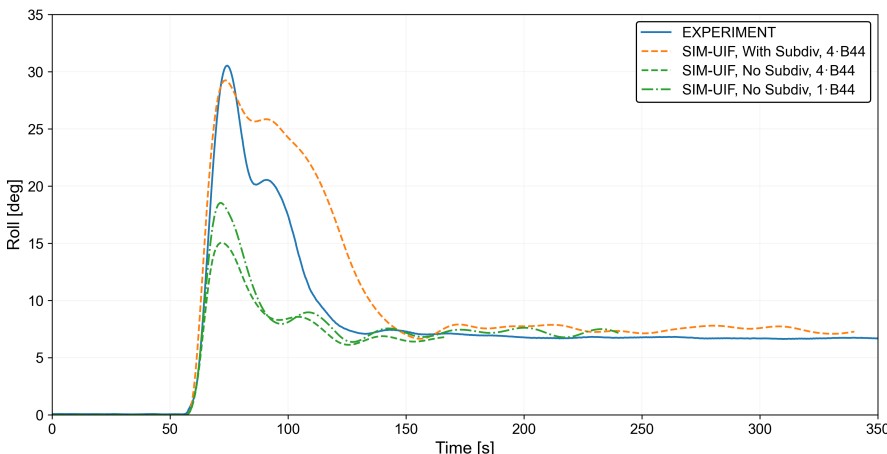

**Figure 16.** Roll response comparison in calm water using the dynamic UIF model.

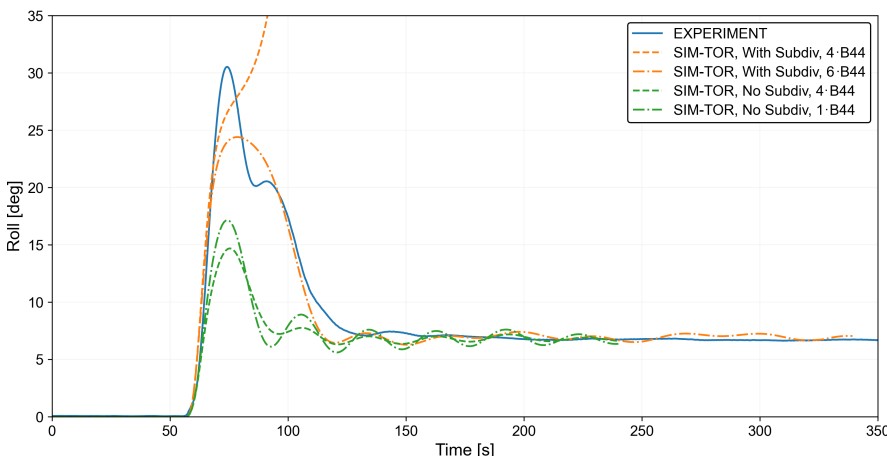

**Figure 17.** Roll response comparison in calm water using the steady Bernoulli-type TOR model.

A comparison of the water levels in the compartments on deck 1 is shown in Figure 18. The compartments on the breached side (ref. REL 1, 2, and 5) rapidly flood at a similar speed to the experiment, but the timing is slightly different. Most likely, there are significant 3D flow effects in the compartment so that the water is unevenly distributed in the compartment. This cannot be captured without further subdivision, which is an interesting research point for future work. On the other hand, once the ship has survived the transient flooding stage, the subdivision on deck 1 is considered less relevant in the progressive flooding phase. The compartments on the intact side (ref. REL 6, 7, and 8) flood in a more complicated manner, which is rather well captured in the aft (Rel 6) and centre (Rel 7) breached compartments, but less good in the most forward compartment (Rel 8). The calculated water level on decks 3, 4, 5, and 6 near the breached side show a fair correlation to the measurements, as shown in Figure 19.

From the results in this section, it is concluded that the global motions and transient flooding of the compartments of the damaged ship can be well captured by the UIF model, given a realistic subdivision of the ship and use of appropriate discharge resistance coefficients. Our simulations show that the roll damping of the damaged ship is important during transient flooding phase and that the roll damping is higher than for the intact ship. The numerical roll damping was tuned against model test experiments, a tuning that might depend on the damage extent, given the reasoning in this section. After the initial transient flooding, the ship might be exposed to progressive flooding in waves with potentially a large amount of flood water on board. The importance of the roll damping during progressive flooding remains a point of research. It is finally concluded that the

steady TOR model can rightfully predict the final flooding condition of the ship in calm water, but that the transient response is predicted less accurately compared to the UIF model using the same level of subdivision.

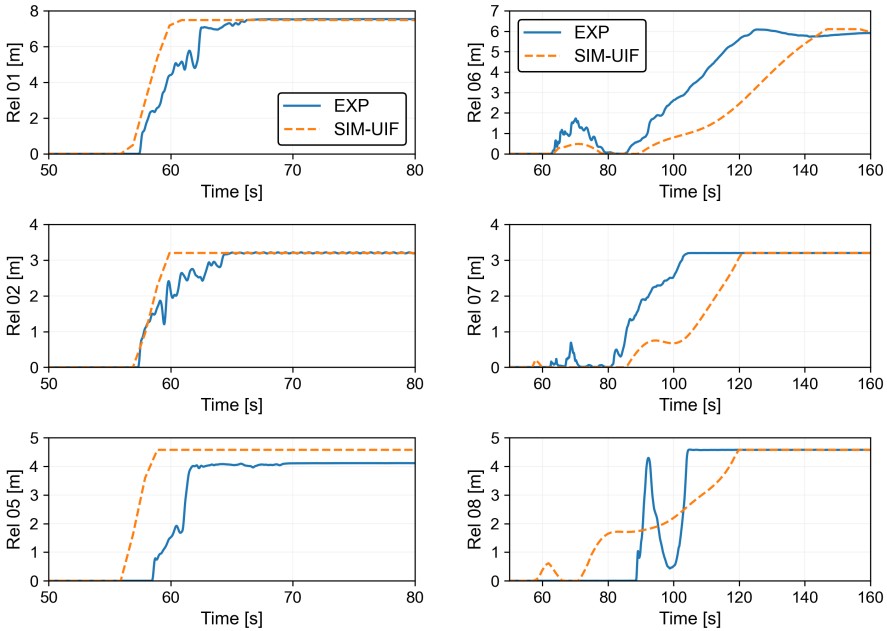

**Figure 18.** Comparison of water level recordings in deck 1 compartments.

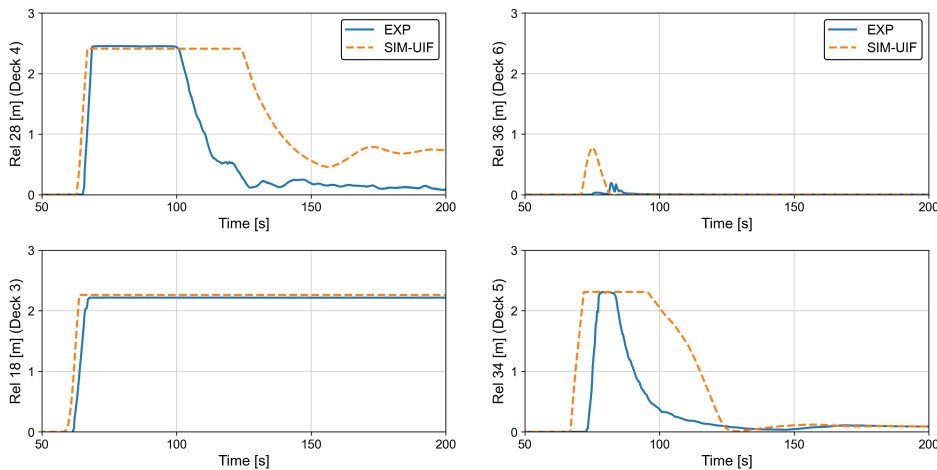

**Figure 19.** Comparison of water level recordings near the forward breach on deck 3, 4, 5, and 6.

## 4. Discussion and Conclusions

The implementation of a new flooding model that includes fluid inertia is presented, and various simulation results are shown and discussed. The new flooding model is implemented in MARIN's XMF time-domain simulation environment and can be used for any flooding configuration onboard ships or for dedicated configurations. Compartments and connecting surfaces are put into a flooding network denoted by the keyword UIF when inertia is modelled, and by TOR when the steady Bernoulli equations are applied.

The first example given is a tank drainage experiment, to measure the discharge coefficient $C_D$ of a rectangular orifice. The discharge coefficient is shown to depend on the Reynolds number and a formulation is presented to capture this effect in simulations. This avoids the user selection of the coefficient and smoothly connects model test and full-scale simulations. Fluid inertia effects do not play an important role in this experiment and were

found to be small in the UIF simulation. The TOR and UIF model are both applicable to model this experiment.

In the second example, a down-flooding experiment is presented between two compartments, of which the top one is breached by water. ComFLOW CFD simulations were performed, showing a very good correlation to the experiments. To obtain a good correlation with the TOR and UIF model, the discharge coefficient for the down-flooding opening had to be set to 0.80. This value could be explained based on the CFD results. Both the TOR and UIF model then predict the down-flooding well. Modelling the air entrapment only slightly changed the UIF results, but showed some realistic free surface oscillations in the modelled venting pipe. It was expected, and confirmed, that the configuration is fully vented, since both compartments have a direct venting opening to the atmosphere, a situation that is not necessarily the case in complex shipflooding scenarios.

In the third example, an oscillating water column (OWC) application is analysed, which represents an energy harvesting device. The modelling of entrapped air is essential in this experiment. The experiment is found to be adiabatic. The UIF solver models an isothermal process, and this explains the difference in the natural period compared to the experiment. The UIF model predicts the dynamics well but the TOR model does not. In this simulation, the time step does not need to be very small for stable and converged results. It is expected that a full scale OWC device behaves as an isothermal process; hence, the UIF model is expected to model a full-scale OWC system well. The fluid inertia effects are well captured by the UIF model.

In the fourth and last example, the flooding of a cruise ship in calm water is presented. The flooding arrangement is complex, in particular in the lowest deck. A logical subdivision is made at points of flow contraction and expansion. The applicable discharge coefficient is calculated to be around 0.8 for such openings, which is different for fluid flowing through an orifice. Using calibrated roll damping coefficients, a very good correlation between the UIF results and the experiments is found, both in global ship roll response and in floodwater distribution through the ship. The results without the deck subdivision converge to the correct list angle, but the transient flooding phase is not well captured. The steady TOR model gives a slightly less accurate prediction of the flooding, requiring a different calibration of the roll damping as well.

The various applications and validations shown demonstrate that the UIF model is a generally applicable fast fluid-solver that models the fluid inertia aspects very well when they are present in the configuration. When the inertia effects are absent or non-governing, the UIF model remains to provide valid results. Further development and research is ongoing with respect to the air entrapment modelling and, in particular, on the robustness side of the coupled fluid–air solver. It will be interesting to generalise the implementation to handle adiabatic processes as well (perhaps in a simplified manner), making the UIF solver even more general. On the other hand, it is recognised that, for shipflooding simulations (including the OWC at full scale), the isothermal air modelling is expected be applicable. This is the main area of application and purpose of the UIF solver.

Further research is ongoing with respect to the flooding of ships at sea in irregular waves, including the associated statistics.

**Author Contributions:** Conceptualisation, R.v.'t.V.; writing, R.v.'t.V., J.v.d.B. and S.B. All authors have read and agreed to the published version of the manuscript.

**Funding:** The research presented in this paper was partly carried out within the framework of the EU research project Flooding Accident Response (FLARE) no. 814753, under H2020 program, funded by the European Union, which is gratefully acknowledged. The UIF flooding model developments were mainly carried out within MARIN's background research program.

**Institutional Review Board Statement:** Not applicable.

**Informed Consent Statement:** Not applicable.

**Data Availability Statement:** EU-FLARE model test data available at http://shipstab.org/index.php/data-access (accessed on 26 April 2023).

**Acknowledgments:** Joris van den Berg is acknowledged as the original developer of the UIF flooding model. Sander Boonstra is acknowledged for the developments of the flooding model within the XMF environment. All three authors were involved in the development, verification, and validation of the flooding model, under the umbrella of ship safety research. All opinions are solely those of the authors.

**Conflicts of Interest:** The authors declare no conflict of interest.

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
