# Peer review of "A Unified Internal Flow Model with Fluid Momentum for General Application in Shipflooding and Beyond"

_jmse, doi:10.3390/jmse11061175_

Round 1

Reviewer 1 Report

The authors introduce an UIF network calculating the flooding between ship compartments using cell-averaged momentum to account for fluid inertia effects. All efforts to model the fluid inertia effects in ship flooding simulations are certainly steps into the right direction. The comparisons between different calculations and with experimental results show that the inertia effect is there, when UIF is applied.

Lee in [9] introduced the dynamic orifice equation (DOE), in which there is an inertia term “added mass” in the DOE related to the area of the damage opening/orifice. When one reads the present paper, it is difficult to find anything similar. Could the authors explain where does UIF inertia term come from? What is the fluid acceleration or change in fluid momentum related to this inertia term?

The cell-averaged momentum has a direction vector? Let us assume we have a flow in a long corridor towards three openings at the end: one straight ahead, and two to both sides. Does the UIF method distribute more water to the compartment straight ahead and less to the compartments at the sides? This takes place in reality. Is the UIF method able to model this? The paper would benefit from more detailed explanations of the UIF modeling itself, in view of the assumptions in the model and the known behavior of the fluid in the compartments.

The authors write about Computational Fluid Dynamics Solver (CFD) as an alternative, which concept includes everything from simple solutions of potential flow with finite differences to RANS, LES and DNS, and still further to meshless Lagrangian schemes SPH etc. Please define this alternative, perhaps you mean RANSE solver, perhaps not?

The lines 18, 58, 85, 126, 153, 239, 291, and 364 may have typographical errors. Please check.

It is not incorrect to write “flood water”, as the authors do, but most other authors write “floodwater” together also in cases of ship flooding.

Author Response

thanks for the comments, appreciated and all included now.

Reviewer 2 Report

This paper treats the dynamics of the flow path for a ship flooding, especially focused on the opening to opening i.e. virtual pipe(the authors terminology). And many simulation results are included with somewhat long explanations.

- Please shorten the content from line 18 to line 61, This paper is for 'a unified internal flow model' not for the 'XMF'. And the amount of pages is too large, It had better to reduce the long explanations.

- Section 2.2. If there are many cells and each cell has openings more than two, the flow path can have multiple flow paths. In that case, the reviewer think that making virtual pipes is too difficult. If there is a limitation of virtual pipe, please describe it in the paper. Or if you have a method, please explain it briefly.

- Equation 6.  p_ambient/rho

- Equation 11.  specify the p_a is the absolute pressure of the air.

Figure 8, Rel 17, the UIF result is differ from the others in time 10-22 sec. Can you explain this?

- Line 438-439. The change of temperature can not cause this difference. So you had batter delete this last paragraph.

- Line 291.  are reported in ?? (please fill in ??)

- Line 511.  opening seen in ?? (please fill in ??)

TYPO

- Line 144.  if a opening (if 'an' opening)

- Line 153.  the nett flow (the 'net' flow)

- Line 364.  At that moment he (modifying to 'the')

- Line 395.  cross ection (it means 'section')

- Line 507. 'than' typo of 'then'?

- Line 600. In the second example (attaching comma ',')

- Line 614. The the third example (deleting 'the' and attaching comma ',')

- Line 623. In the fourth and last example (attaching comma ',')

Author Response

(The authors gave the same response as above.)
